# Antiangiogenic Effect of Dopamine and Dopaminergic Agonists as an Adjuvant Therapeutic Option in the Treatment of Cancer, Endometriosis, and Osteoarthritis

**DOI:** 10.3390/ijms241210199

**Published:** 2023-06-15

**Authors:** Julieta Griselda Mendoza-Torreblanca, Noemi Cárdenas-Rodríguez, Jazmín Carro-Rodríguez, Itzel Jatziri Contreras-García, David Garciadiego-Cázares, Daniel Ortega-Cuellar, Valentín Martínez-López, Alfonso Alfaro-Rodríguez, Alberto Nayib Evia-Ramírez, Iván Ignacio-Mejía, Marco Antonio Vargas-Hernández, Cindy Bandala

**Affiliations:** 1Laboratorio de Neurociencias, Subdirección de Medicina Experimental, Instituto Nacional de Pediatría, Mexico City 04530, Mexico; 2Laboratorio de Medicina Traslacional Aplicada a Neurociencias, Enfermedades Crónicas y Emergentes, Escuela Superior de Medicina, Instituto Politécnico Nacional, Mexico City 11340, Mexico; 3Laboratorio de Biología de la Reproducción, Subdirección de Medicina Experimental, Instituto Nacional de Pediatría, Mexico City 04530, Mexico; 4Unidad de Ingeniería de Tejidos, Terapia Celular y Medicina Regenerativa, Instituto Nacional de Rehabilitación Luis Guillermo Ibarra Ibarra, Mexico City 14389, Mexico; 5Laboratorio Nutrición Experimental, Instituto Nacional de Pediatría, Secretaría de Salud, Mexico City 04530, Mexico; 6Neurociencias Básicas, Instituto Nacional de Rehabilitación Luis Guillermo Ibarra Ibarra, Secretaría de Salud, Mexico City 14389, Mexico; 7Servicio de Reconstrucción Articular, Cadera y Rodilla, Instituto Nacional de Rehabilitación Luis Guillermo Ibarra Ibarra, Mexico City 14389, Mexico; 8Laboratorio de Medicina Traslacional, Escuela Militar de Graduados de Sanidad, Mexico City 11200, Mexico; 9Subdirección de Investigación, Escuela Militar de Graduados de Sanidad, Mexico City 11200, Mexico

**Keywords:** antiangiogenic, cancer, dopamine (DA), dopamine agonists (DA-Ag), endometriosis, osteoarthritis (OA), vascular endothelial growth factor (VEGF), vascular endothelial growth factor receptor (VEGFR)

## Abstract

Dopamine (DA) and dopamine agonists (DA-Ag) have shown antiangiogenic potential through the vascular endothelial growth factor (VEGF) pathway. They inhibit VEGF and VEGF receptor 2 (VEGFR 2) functions through the dopamine receptor D2 (D2R), preventing important angiogenesis-related processes such as proliferation, migration, and vascular permeability. However, few studies have demonstrated the antiangiogenic mechanism and efficacy of DA and DA-Ag in diseases such as cancer, endometriosis, and osteoarthritis (OA). Therefore, the objective of this review was to describe the mechanisms of the antiangiogenic action of the DA-D2R/VEGF-VEGFR 2 system and to compile related findings from experimental studies and clinical trials on cancer, endometriosis, and OA. Advanced searches were performed in PubMed, Web of Science, SciFinder, ProQuest, EBSCO, Scopus, Science Direct, Google Scholar, PubChem, NCBI Bookshelf, DrugBank, livertox, and Clinical Trials. Articles explaining the antiangiogenic effect of DA and DA-Ag in research articles, meta-analyses, books, reviews, databases, and clinical trials were considered. DA and DA-Ag have an antiangiogenic effect that could reinforce the treatment of diseases that do not yet have a fully curative treatment, such as cancer, endometriosis, and OA. In addition, DA and DA-Ag could present advantages over other angiogenic inhibitors, such as monoclonal antibodies.

## 1. Introduction

Dopamine (DA) and dopamine agonists (DA-Ag) are known for their therapeutic effects in diseases involving neurochemical alterations in the nervous system [1]. However, these compounds have different biochemical properties that allow them to be applied to treat other diseases, as is the case with their antiangiogenic effect, a property that can be applied to treat pathologies where angiogenesis is an important physiological mechanism, such as cancer [2], endometriosis [3], and osteoarthritis (OA) [4].

The formation of blood vessels, especially in cancer, can occur through different mechanisms such as germinative angiogenesis, intussusceptive angiogenesis, vasculogenesis, endothelial progenitor cells recruitment, vascular mimicry, and cancer stem cells transdifferentiation. The vascular endothelial growth factor (VEGF)/VEGF receptor 2 (VEGFR 2) pathway is involved in all these mechanisms [5,6]. Different antiangiogenic strategies have been proposed; one of them is DA and DA-Ag since their effect has been related to the inhibition of the VEGF/VEGFR 2 pathway, where it has been observed that these compounds showed antiproliferative effects by blocking the formation of blood vessels in tumors [7], decreasing endometriotic lesions [8,9,10], and decreasing inflammation and apoptosis in cartilage degeneration and bone sclerosis [11,12,13]. DA and DA-Ag inhibit VEGF/VEGFR 2 functions through the dopamine receptor D2 (D2R), which promotes VEGFR 2 endocytosis and prevents VEGFR 2 phosphorylation [14,15,16] and decreases the main proangiogenic stimulus (VEGF binding to VEGFR 2). With this knowledge, clinical trials are currently underway to test different DA-related drugs to evaluate their efficacy as antiangiogenic options in both related and unrelated diseases of the nervous system.

Moreover, although there are other antiangiogenic agents targeting the VEGF/VEGFR 2 pathway, especially monoclonal antibody inhibitors of VEGF/VEGFR 2, DA and DA-Ag have several advantages, as they are more economically accessible and have greater availability and probably fewer adverse effects [17,18,19]. Therefore, the purpose of this review is to describe the mechanisms associated with the angiogenesis of DA and DA-Ag in cancer, endometriosis, and OA through the analysis of experimental and clinical studies that show their potential as adjuvants. In addition, we analyzed the advantages and disadvantages of DA and DA-Ag compared with VEGF/VEGFR 2 monoclonal antibodies. Although the antiangiogenic mechanism mediated by the VEGF/VEGFR 2 pathway is one of the most important mechanisms [5,6], we should not omit that alternative proangiogenic pathways could be activated [6], especially in cancer and possibly in endometriosis and OA as an adaptive response. Further studies are needed to describe whether DA and DA-Ag treatment could activate the alternative proangiogenic mechanism. Therefore, the study of DA and DA-Ag as an antiangiogenic combined with other alternative proangiogenic pathway inhibitors could be an effective antiangiogenic treatment.

## 2. Methodology

For the selection and analysis of the articles included in this review, the following databases were consulted: PubMed, Web of Science, SciFinder, ProQuest, EBSCO, Scopus, Science Direct, Google Scholar, PubChem, NCBI Bookshelf, DrugBank, livertox, and Clinical Trials. Articles on DA, DA-Ag, and its receptors, as well as those postulating or demonstrating their antiangiogenic effect in any phase of study, particularly in cancer, endometriosis, and osteoarthritis, were considered. Original manuscripts, reviews, minireviews, systematic reviews, meta-analyses, clinical trials, books, and specialized databases were included. The search was performed by applying the following keywords alone or in combination: “dopamine”, “dopaminergic drug”, “chemical compounds”, “chemical structure”, “DA receptors”, “precursors”, “experimental DA-Ag and antagonists”, “receptor blockers”, “cancer”, “endometriosis”, “osteoarthritis”, and “drug repositioning”. Finally, a total of 203 references were considered. A total of 203 bibliographic sources from 1972 to 2023 were obtained.

## 3. Dopamine (DA) and Its Receptors

DA is a key neuromodulator that is synthesized in both the central nervous system (CNS) and the peripheral nervous system [20,21]. In the brain, DA is involved in the regulation of executive functions, including locomotor activity, cognition, emotion, arousal, reward, sexual behavior, and lactation [22]. Recently, it has been shown to play a critical role in metabolic regulation [23]. DA exerts its actions through the activation of G protein-coupled receptors. Due to the relevant functions of DA, the dysregulation of dopaminergic signaling is implicated in the development of several pathologies, such as Parkinson’s disease, Huntington’s disease, schizophrenia, attention deficit and hyperactivity disorder, and addiction [22]. In the late 1980s, Bunzow et al. cloned the first five DA receptor proteins [24] and classified them according to their pharmacological, biochemical, and physiological function as D1-like (D1R, D5R) and D2-like (D2R, D3R, and D4R) and showed their expression in a wide range of structures within the CNS [25]. They are encoded by different genes located at different chromosomal loci and show considerable homology in their protein structure and function. The D1R and D5R receptors are encoded by genes with no introns and share 80% of their identity, whereas D2R and D3R share 75% and 3% of their identity in the D4R transmembrane domains, respectively. The last three receptor subtypes are encoded by genes with introns. Among all the DA receptors, D1R and D2R are the most abundant in the CNS; specifically, they are found in the medium spiny striatum neurons and hippocampus [26,27,28,29]. DA receptors are integral membrane proteins coupled to G proteins (Gs) with seven membrane-spanning α-helical domains to maintain the three-dimensional structure of the receptor [30].

Functionally, DA receptors mediate the effects of DA and dopaminergic compounds through different signaling pathways; thus, upon activation by dopaminergic compounds, D1R activates adenylyl cyclase (AC), which in turn increases the amount of the second messenger cyclic AMP (cAMP). In contrast, dopamine D2-like receptors (D2R, D3R, and D4R) have an inhibitory effect on adenylyl cyclase, leading to a decrease in cAMP levels [31,32]. This signaling is mediated by different G proteins (Gs), mainly Gsα for the stimulation and Giα for the inhibition of AC [33,34]. In addition to the G protein-dependent pathway, other signaling pathways are utilized by D1R; thus, after activation by DA, D1R promotes the accumulation of β-arrestin2 protein that is subsequently internalized by endocytosis and concomitantly results in a loss of cell surface receptors and additional arrestin-mediated signaling events [35]. Interestingly, altered D1R and D2R signaling has been associated with many neurological diseases, such as schizophrenia, Parkinson’s disease, attention deficit hyperactivity disorder, and autism [36,37,38,39]. Due to the relevance of dopaminergic signaling, many drugs that are targets of D1R and D2R have been developed to combat various CNS diseases through the modulation of the dopaminergic system, whose homeostasis is indispensable for the treatment of various CNS disorders by maintaining normal dopaminergic homeostasis and restoring homeostasis in disease states.

## 4. Dopamine (DA) and Dopamine Agonists (DA-Ag)

DA synthesis begins with the hydroxylation of L-tyrosine by the tyrosine hydroxylase enzyme (TH) to generate L-3,4-dihydroxyphenylalanine (L-DOPA); then, aromatic L-amino acid decarboxylase (DOPA decarboxylase) enables cytosolic DA production. DA is released into the synaptic cleft and then recycled and degraded by enzymes for catabolism [34,40,41]. In glial cells, monoamine oxidase (MAO) breaks down DA, producing 3,4-dihydroxyphenylacetaldehyde (DOPAL); in turn, aldehyde dehydrogenase (ALDH) converts DOPAL to carboxylic acid 3,4-dihydroxyphenylacetic acid (DOPAC), or alcohol dehydrogenase (ADH) reduces DOPAL to 3,4-dihydroxyphenylethanol (DOPET) [34,41,42]. The enzyme catechol O-methyl-transferase (COMT) catalyzes the methylation of DA to 3-methoxytyramine (3-MT), which is a MAO substrate that forms 3-methoxy-4-hydroxyphenylacetaldehyde (HMPAL). Finally, the enzyme ALDH catalyzes HMPAL to generate homovanillic acid (HVA), the major end-product of DA degradation [34,41,42]. DA binds to D1-like and D2-like receptors, which are G-protein-coupled channels. The D1-like receptor increases protein phosphorylation, whereas the D2-like receptor decreases protein phosphorylation [34,43,44,45].

Regarding the role of DA as a therapeutic agent, the literature has shown the use of DA precursors, agonists, antagonists, and blockers as therapeutic agents in some diseases. DA precursors such as L-DOPA (levodopa) have been used for Parkinson’s disease, and L-phenylalanine and L-tyrosine have been used as antidepressants [46,47,48]. DA-Ag, apomorphine, bromocriptine, cabergoline, lisuride, piribedil, pramipexole, ropirinole, and rotigotine have been used as anti-Parkinsonians [49,50,51,52,53,54,55]. In addition, ropinirole and rotigotine have been used for restless legs syndrome [56,57]; cabergoline and quinagolide have been used for hyperprolactinemia [50,52]; bromocriptine has been used for amenorrhea, galactorrhea, and female infertility [52]; and DA and fenoldopam have been used for hemodynamic imbalances [52,58]. Experimental agonists such as LS-186,899, pukateine, quinpirole, and SKF 38393 are currently being tested in different scientific studies [34,59,60,61,62,63,64,65].

## 5. Antiangiogenic Capacity of Dopamine (DA) and Dopamine Agonists (DA-Ag) and the Mechanisms of Action

Angiogenesis, the generation of new blood vessels from the existing vasculature, plays a key role in physiological processes such as embryonic development, wound healing, and organ regeneration, as well as in various pathologies, such as cancer, diabetes, retinopathies, and tumor metastasis [15,66]. Several molecular mechanisms have been explored to understand the basic processes underlying angiogenesis. Signaling mediated by VEGF and its target receptors has been identified as an important player in angiogenesis and vascular permeability, among others [66,67]. The VEGF family genes are composed of five members, including VEGF-A (VEGF), VEGF-B, VEGF-D, and placental growth factor [68,69,70,71], whereas VEGFR is represented by three members, namely, VEGFR 1, VEGFR 2, and VEGFR 3, where VEGFR 2 is the main regulator of physiological and pathological angiogenesis [15,72,73]. The signal transduction cascade begins when VEGF binds to VEGFR 2, leading to a conformational change, dimerization, and phosphorylation of tyrosine residues of the receptor, which leads to the activation of several intracellular molecules that serve as downstream signaling elements that propagate the signal to activate angiogenesis [72]. This system mediates angiogenesis through the proliferation, migration, and survival of endothelial cells, promoting new vessel formation (Figure 1) [74,75]. The potent proangiogenic activity of VEGF was first described as essential for vascular endothelial cells; however, VEGF and VEGF receptors are expressed on numerous nonendothelial cells, including tumor cells [72,76]. In addition, VEGFR 2 is associated with the mitogenic, angiogenic, and permeability-enhancing effects of VEGF in a wide variety of tissues [75,77].

DA and DA-Ag (e.g., bromocriptine, cabergoline, quinagolide, and quinpirole) have demonstrated antiangiogenic properties in different pathologies [14,78,79,80,81,82]. Sarkar et al. demonstrated that DA administered intraperitoneally (50 mg/kg/day) in mice with colon cancer was able to inhibit angiogenesis and tumor growth without apparent adverse effects [83]. This antiangiogenic capacity has been associated with the VEGF pathway [66,67,83], and several mechanisms have been described. For example, Basu et al. reported that DA and its D2R DA-Ag, bromocriptine, and quinpirole inhibited mouse ovarian tumor-induced angiogenesis and inhibited human umbilical vein endothelial cell proliferation and migration [14]. The authors described, for the first time, the antiangiogenic relationship of the DA-D2R/VEGF-VEGFR 2 mechanism, with the induction of VEGFR 2 endocytosis being the key act for the arrest of angiogenesis by DA [14]. Indeed, the internalization and inactivation of VEGFR 2 downregulate several proangiogenic factors and upregulate antiangiogenic factors, resulting in an unstructured blood supply (Figure 2) [80].

Another D2R-related antiangiogenic mechanism was observed in tumor and normal endothelial cells. Normal endothelial cells show very low or no expression of DA-D2R compared to tumor endothelial cells [14,84]. Through paracrine signaling, VEGF secreted by tumor cells can stimulate D2R expression by activating the extracellular-signal-regulated kinase1/2 (ERK1/2) signaling cascade and increasing Krüppel-like factor 11 (KLF11) expression in endothelial cells (Figure 2) [79]. Increased D2R can inhibit VEGF-induced angiogenesis [14,84,85] as a feedback mechanism that regulates the actions of VEGF on endothelial cells [79].

Furthermore, through its D2R, DA not only has an effect on decreasing angiogenesis [86,87], but it also inhibits tumor endothelial cell proliferation through the inactivation of VEGF-induced mitogen-activated protein kinase (MAPK) and focal adhesion kinase (FAK) phosphorylation (Figure 2) [78,88]. FAK is a tyrosine kinase that promotes p53 degradation via ubiquitination, leading to tumor cell growth and proliferation, angiogenesis, and vascular permeability [89]. In addition, D2R receptors can decrease matrix metalloprotease (MMP-9) (ERK1/2-mediated) release by endothelial progenitor cells, inhibiting their mobilization from the bone marrow and preventing their participation in tumor neovascularization [88,90,91].

Moreover, it has been reported that DA can inhibit VEGF-induced endothelial cell migration. By acting through D2R, DA can regulate the phosphorylation of different tyrosine residues of VEGFR 2, leading to the inactivation of different downstream signaling pathways [15,92,93]. Sinha et al., 2009, in isolated human umbilical cord endothelial cells, demonstrated that treatment with 10 μM DA prior to VEGF stimulation at 10 ng/mL produced an increase in the VEGF-induced phosphorylation of phosphatase-2 containing Src homology region 2 domain (SHP-2) and its phosphatase activity. Active SHP-2 dephosphorylates VEGFR 2 at Y951, Y996, and Y1059 but not at Y1175 (15). The decreased phosphorylation of VEGFR 2 at Y951 leads to a subsequent decrease in Src phosphorylation at Y418 and its kinase activity, inhibiting cell migration (Figure 2) [15]. SHP-2 knockdown was also observed to affect the DA-regulated inhibition of VEGF-induced VEGFR 2 phosphorylation and, subsequently, the activation of Src, a protein related to cancer progression [15,94].

Figure 2 shows an integrated and simplified scheme of the main antiangiogenic mechanisms of the DA-D2R/VEGF-VEGFR 2 system in different pathologies. These mechanisms can occur in isolation or together in different cells, organs, or diseases. Future research should be carried out to clarify this question.

**Figure 2 ijms-24-10199-f002:**
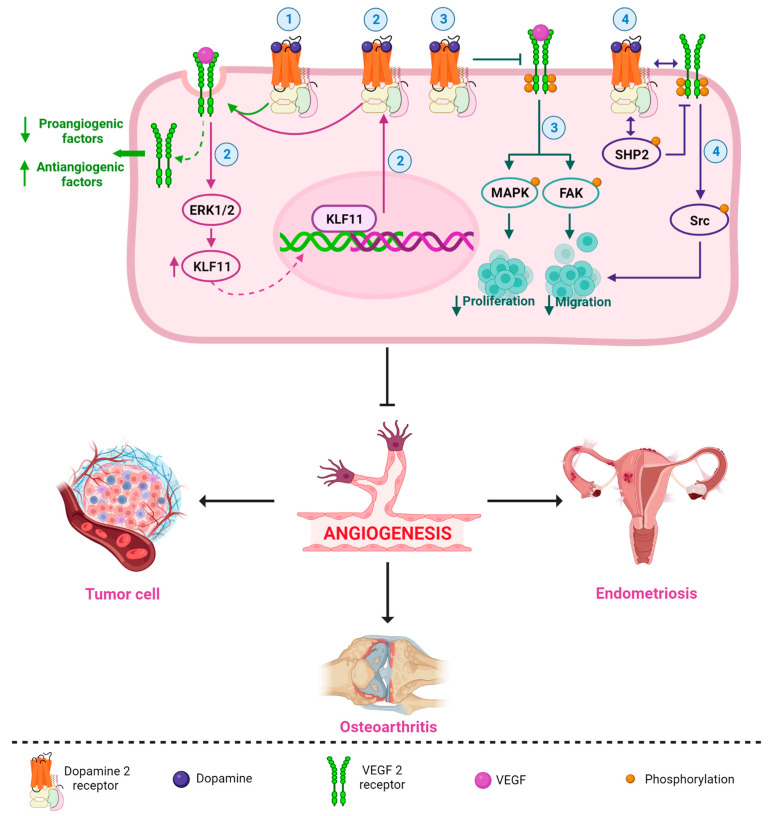
Simplified and integrated scheme of the main antiangiogenic mechanisms described for the DA-D2R/VEGF-VEGFR 2 system in different pathologies. DA (and its DA-Ag, such as bromocriptine, cabergoline, and quinagolide) strongly and selectively inhibit VEGF/VEGFR 2 functions; for example, (1) DA/D2R promote the induction of VEGFR 2 endocytosis, which reduces a series of proangiogenic factors and increases antiangiogenic factors [14,80]; (2) VEGF can stimulate D2R expression by activating the ERK1/2 signaling cascade and upregulating KLF transcription factor 11 expression, and upregulation of D2R can inhibit VEGF-induced angiogenesis (perhaps by promoting VEGFR 2 endocytosis) [79]; (3) DA/D2R inhibits VEGF-induced activation of MAPK and FAK phosphorylation, blocking cell proliferation and migration [88]; and (4) DA may regulate dephosphorylation of different tyrosine residues of VEGFR 2, leading to inactivation of different downstream signaling pathways. DA causes an increased association of D2R with VEGFR 2. DA also induces an increase in the association between SHP-2 (a protein phosphatase) and D2R and stimulates the phosphorylation of SHP-2. Then, active SHP-2 inhibits the phosphorylation of VEGFR 2. The decrease in VEGFR 2 phosphorylation leads to a subsequent decrease in Src phosphorylation, blocking VEGF-induced migration [15]. All these mechanisms (separately or together) may be involved in the inhibition of angiogenesis in different diseases, such as endometriosis, cancer, and osteoarthritis.

## 6. Therapeutic Potential of Dopamine (DA) and Dopamine Agonists (DA-Ag) as Antiangiogenic Agents

As mentioned above, cancer, endometriosis, and OA are diseases in which angiogenesis is an important physiological mechanism; therefore, the development and maintenance of these conditions can be affected by antiangiogenic therapy, a promising strategy that in the last decade has had an increasing number of studies in which different antiangiogenic agents have been used to inhibit tumor growth, induce the regression of endometriotic lesions, and inhibit osteogenesis by targeting their blood supply [83,84,85,95,96,97]. Drugs related to the VEGF/VEGFR 2 signaling pathway, phytochemicals, immunomodulators, antihormonal drugs, and DA-Ag have been used for their antiangiogenic capacity [3,80,81,98]. Of relevance has been the use of monoclonal antibodies targeting the VEGF pathway for cancer therapy, where studies have shown them to be key inhibitors of tumor angiogenesis during adjuvant, maintenance, or combination therapy against some solid tumors [99,100]. However, not all these compounds have demonstrated safety or tolerability; therefore, it would be important to test antiangiogenic treatments or adjunctive treatments for endometriosis, cancer, or OA with compounds that have been shown to have a favorable safety profile and are already clinically approved for the treatment of other diseases [80,101]. In this context, since DA and DA-Ag have an acceptable safety profile and are clinically approved, they may represent an alternative to many antiangiogenic agents.

### 6.1. Therapeutic Potential of Dopamine and Dopamine Agonists in Endometriosis

Endometriosis is a common gynecologic disease characterized by the presence of endometrial tissue, glands, and stroma outside the uterine cavity and is a common estrogen-dependent disorder associated with pelvic pain and infertility. Its etiology is unknown, and treatment is surgical with a high risk of recurrence [81,95,102]. Although many aspects of the pathogenesis of endometriosis are not fully established, endometriotic lesions grow in areas with a constant and abundant blood supply, and angiogenesis is a prerequisite for the invasion, proliferation, long-term growth, and maintenance of ectopic implants [81,98,103]. Under this rationale, the use of commercial antiangiogenic drugs has been explored in preclinical models of endometriosis; however, endometriosis specifically affects women of reproductive age, and the selection of antiangiogenic agents is very important, as physiological angiogenic processes such as follicle maturation, corpus luteum function, eutopic endometrial proliferation, and embryo development must be carefully protected [98,101].

DA and DA-Ag have shown a benign clinical profile and several advantages for women with endometriosis, as they are already used for hyperprolactinemia and lactation suppression, do not seem to interfere with physiologic angiogenesis in reproductive organs, and do not interfere with ovulation and spontaneous pregnancy [3,80,96]. The ergot cabergoline and bromocriptine and the nonergot quinagolide are the main D2R DA-Ag tested in different preclinical and clinical studies [82,104,105]. In experimental models of endometriosis, these DA-Ag have been shown to downregulate a series of proangiogenic factors and upregulate antiangiogenic factors in inflammatory, endothelial, and endometrial cells, targeting the newly formed and mature vasculature and resulting in an unstructured blood supply and reduction in lesion size [80]. Table 1 shows the different experiments and doses used by multiple authors, as well as their results regarding their effect on endometriosis in animals and humans.

### 6.2. Therapeutic Potential of Dopamine and Dopamine Agonists in Cancer

It has been shown that D2R is upregulated in many cancers, and the use of D2R DA-Ag has an anticancer efficacy. The protein and gene expression of D2R was observed in patient samples or cell lines with different types of breast, cervical, brain, and lung cancers. The use of D2R DA-Ag affects different metabolic processes, including autophagy, apoptosis, survival signaling, and proliferation, showing that the use of these drugs as anticancer agents might have chemotherapeutic utility [2]. In relation to the angiogenesis process, it is well established that DA-Ag decreases tumor angiogenesis by inhibiting VEGFR 2-mediated signaling in endothelial cells. Previous studies have shown that DA inhibits the proliferation and migration of the VEGF-induced endothelial cell line HUVEC [117,118,119,120]. The D2R agonist stopped the growth of lung cancer in a human xenograft model, and some of the beneficial antiangiogenic effects of D2R DA-Ag may occur through the inhibition of nicotinamide adenine dinucleotide phosphate (NADPH) oxidase (responsible for producing reactive oxygen species) since it promotes angiogenesis [86]. Regarding catecholamine tumor studies, DA, by acting through D2R, inhibits angiogenesis by suppressing the action of the vascular permeability factor and VEGF [121] in both adult endothelial cells and endothelial progenitor cells. In contrast, norepinephrine and epinephrine, by acting through β-adrenoceptors, promote the synthesis of proangiogenic factors in tumor cells [122]. Angiogenesis is related to tumor growth. In aggressive cancer, the blood supply is increased, and endothelial epinephrine cells mobilize from the bone marrow to the tumor site. DA in different kinds of cancer reduces both angiogenic mechanisms [87,123]. Subsequently, further studies with DA-Ag showed the capability to increase D2R expression in endothelial cells, promoting the internalization of VEGFR 2 (Figure 2). Endothelial cells and macrophages reduce VEGF expression and release into peritoneal fluid. In addition, the availability of plasminogen activator inhibitor-1 decreases, which improves fibrinolysis and diminishes angiogenesis [80]. The antiangiogenic activity of cabergoline is expressed in two ways. First, the interaction of cabergoline with D2Rs results in a reduction in prolactin (PLR) cell function, causing a general and local decrease in PRL levels [124]. Cabergoline causes a decrease in PRL, leading to hemoxygenase-1-dependent angiogenesis in macrophages due to decreased VEGF levels. Second, the interaction of cabergoline with D2R leads to the disruption of VEGF binding to its receptor VEGFR 2 and the blockage of VEGF and VEGFR 2 transcription, resulting in an antiangiogenic effect [112]. The antiangiogenic effect of DA and DA-Ag has been demonstrated in vitro and in vivo (Table 2) in different types of cancer; however, more research is still required in this regard.

### 6.3. Therapeutic Potential of Dopamine and Dopamine Agonists in Osteoarthritis

One of the fundamental characteristics of OA is the wear and tear of articular cartilage. During this process, the reactivation of chondrocyte maturation toward hypertrophy occurs, resulting in bone formation at the edges of the articular surface (osteophytes) [134,135,136]. Articular cartilage maintains a stable phenotype throughout life; however, with aging or articular cartilage injury, OA appears. Healthy articular cartilage lacks nerve endings and blood vessels, which form simultaneously with bone formation during OA [137,138]. Because of the formation of these nerve endings, OA is a chronic pain condition; however, the role of neurotransmitters during OA is just beginning to be understood [139,140,141,142,143,144]. Although cartilage lacks nerve endings, chondrocytes have been found to express some catecholamine receptors, and the role of DA during bone formation has been studied in vitro and also in fracture models in adult individuals [140,145,146]. Another important aspect in the pathogenesis of OA is inflammation, which is exacerbated in the initial phase of OA, mainly by cytokines such as interleukin-1β (IL-1β) and tumor necrosis factor alpha (TNFα), which can also contribute to the damage of articular chondrocytes and ultimately to the dedifferentiation of chondrocytes to a fibrocartilaginous phenotype and finally to bone formation [144,147,148,149].

DA has an anti-inflammatory effect; it has been shown to inhibit the production of proinflammatory cytokines such as interleukin-6 (IL-6), TNFα, and inducible nitric oxide synthase (iNOS) induced by lipopolysaccharide in microglia, and this appears to occur through blocking nuclear factor kappa-light-chain-enhancer of activated B cells (NF-κB) signaling [150]. This led us to believe that the role of DA in OA would have a “protective” role on articular cartilage damage in vivo and in vitro models of OA. In a study performed in an in vitro experimental model of OA that consisted of treating chondrocytes with IL-1β, it was shown that in DA-treated chondrocytes, they also upregulate the NF-κB and Janus kinase/signal transducer and activator of transcription 3 (JAK/STAT3) signaling pathways, inhibiting their nuclear activation and leading to the inhibition of articular cartilage damage markers such as iNOS, cyclooxygenase-2 (COX-2), metalloproteases-1, -3 and -13 (MMP-1, MMP-3, and MMP-13), while markers of healthy articular cartilage such as type II collagen and glycosaminoglycan content were downregulated [139]. In the same work, they found that in an animal model of OA, in mice with destabilization of the medial meniscus, when treated with DA, the effects of joint damage were reversed.

Articular cartilage is an avascular tissue, but in OA, when the synovial membrane becomes inflamed, angiogenesis begins, which is the appearance of blood vessels in the articular cartilage, accelerating the process of bone formation on the cartilage surface with subsequent osteophyte formation [151,152]. Angiogenesis can be caused by synovitis from inflammatory cells such as macrophages that secrete VEGF [153] and in turn induce other cell types such as endothelial cells and fibroblasts to secrete other angiogenic factors such as TNFα and IL-1β [154,155]. On the other hand, angiopoietin plays a regulatory role in angiogenesis by controlling cartilage vascularization [156], and this angiopoietin under normal conditions is produced by synoviocytes. Cartilage degradation accompanies pannus formation and is regulated by the activity of MMP-3, MMP-9, and MMP-13 [157,158,159,160,161], which promote the IL-1β-promoted turnover of the extracellular matrix [149,162,163,164]. Under normal conditions, articular chondrocytes produce antiangiogenic factors such as Troponin-1 and Chondromodulin-1, among others, as well as inhibitors of metalloproteinase [159,165,166,167]. In contrast, hypertrophic chondrocytes present receptors for angiogenic factors, which initiates the last phase of the endochondral ossification in long bones, but in joints, this only occurs when chondrocytes are damaged [166,168]. Another important factor regulating angiogenesis is hypoxia-inducible factor (HIF-1), which joint chondrocytes need to survive in a hostile environment with low oxygen levels (hypoxia) [169,170]; if these levels increase, then the transcription factor SRY-box transcription factor 9 (SOX-9) decreases its expression, and chondrocytes rapidly mature into hypertrophic chondrocytes [171,172], which is an important step in endochondral ossification but also in the establishment of OA. Additionally, the adipokine visfatin has been reported to increase VEGF-dependent angiogenesis, and patients with OA have elevated visfatin levels [173].

The regulation of angiogenesis is thus one of the important points for the treatment of OA. Thus, VEGF signaling may be one of the main therapeutic targets for this disease. The inhibition of VEGF signaling can reduce the progression of OA, and the use of bevacizumab, which is an antibody against VEGF, inhibited angiogenesis and the progression of OA in a rabbit model of OA and increased the thickness and quality of articular cartilage [174,175]. In addition, how miR-485-5p, which is the shRNA of visfatin, inhibits angiogenesis and OA progression in a rat model of OA has been studied [173]. All this creates opportunities for researchers to design treatments that block angiogenesis in OA patients and reduce articular cartilage damage.

It is possible that during OA, DA may act on stem cells that contribute to osteophyte formation during OA, as several papers show how DA inhibits mesenchymal stem cell migration through its D2R [176] and may contribute to bone mass loss and inhibit osteogenesis [97], as it has also been shown to suppress rat bone marrow stem cell differentiation through the protein kinase B/Glycogen synthase kinase-3 beta/β-catenin (AKT/GSK-3β/β-catenin) pathway. Thus, the modulation of DA receptors in osteoblasts has also been proposed as a possible therapy to induce healing in those with rheumatoid arthritis [140] and possibly osteoporosis. We think that, in contrast, the activation or application of DA can regulate the wear and deterioration of articular cartilage in different mechanisms, such as by regulating inflammation, controlling chondrocyte maturation toward hypertrophy, and consequently inhibiting osteoblast formation on the articular surface, where osteophytes are usually formed from fibrochondrocytes or bone marrow stem cells; additionally, we think that DA can reduce angiogenesis through D2R, which suggests that using DA can be an important antiangiogenic strategy to treat OA, but further studies are needed to clarify this issue.

Another indication that DA contributes to cartilage maintenance may be due to a reciprocal role in the sonic hedgehog (Shh) signaling pathway, as it is well known that Shh is required for dopaminergic neuronal development [177,178], rat bone marrow mesenchymal stem cells express dopaminergic genes, and Indian hedgehogs control chondrocyte differentiation and maturation during skeletogenesis. However, the role of the hedgehog pathway is controversial; although there is much evidence that the overactivation of the pathway leads to OA pathogenesis, the complete abrogation of the pathway also results in the same problem [179,180,181,182].

Although there are indications that DA may help reduce articular cartilage wear and reduce pain, few treatments for OA are being applied. One study proposed crosslinked hyaluronic acid infiltration with DA to improve joint lubrication and repair articular cartilage [4]. However, studies have not focused on demonstrating the impact of DA on the inhibition of angiogenesis in OA. Table 3 shows some studies that have evaluated DA in OA.

## 7. Advantages and Disadvantages of Dopamine (DA) and Dopamine Agonists (DA-Ag) Compared to Monoclonal Antibody Inhibitors of the VEGF/VEGFR 2 Pathway

Traditionally, antiangiogenic agents targeting the VEGF signaling pathway can be broadly divided into three categories: (1) anti-VEGF antibodies, (2) anti-VEGFR antibodies, and (3) VEGFR tyrosine kinase inhibitors (TKIs) [18,19]. Monoclonal antibody-based therapy is one of the most important strategies used to treat patients with various diseases. To date, since 1975, at least 570 therapeutic monoclonal antibodies have been studied in clinical trials by commercial companies, 79 therapies have been approved by the U.S. Food and Drug Administration and are currently on the market, and many more are being evaluated in clinical trials [187]. Antiangiogenesis monoclonal therapy began in 2004 with the approval of bevacizumab (avastin), a humanized anti-VEGF-A monoclonal antibody that acts by selectively binding to circulating VEGF and thereby inhibits the binding of VEGF to its cell surface receptors [188]. Other examples of VEGF-A antagonist monoclonal antibodies include ranibizumab and brolucizumab [187]. Moreover, ramucirumab (cyramza) and tanibirumab were developed as direct VEGFR 2 antagonists that target VEGFR 2 and block the binding of multiple VEGF ligands to that receptor [18,19,189,190]. In addition to monoclonal antibodies, TKIs were developed to block the kinase activity of VEGFRs and their downstream signal transduction to suppress endothelial proliferation and disrupt vascular nutrient and oxygen supply [19]. Several TKIs have been approved, including sunitinib, sorafenib, pazopanib, vandetanib, axitinib, cediranib, vatalanib, motesanib, regorafenib, cabozantinib, and lenvatinib [19,188,189]. Additionally, a second generation of multikinase inhibitors with an improved target affinity, better toxicity profiles, and fewer off-target effects have been developed [19].

Many biotech companies have heralded antiangiogenic monoclonal antibodies and TIKs as “magic bullets”; however, two major drawbacks of these therapies must be considered: side effects and cost [17,83]. Although antiangiogenesis strategies have been effective at suppressing tumor progression and metastasis in combination with chemotherapy for years, an increased number of adverse events have been associated with them; bleeding, clots that can lead to a stroke or myocardial infarction, arterial hypertension, proteinuria, and gastrointestinal disorders have been reported [17]. Additionally, hand–foot syndrome, diarrhea, and gastrointestinal perforation were significantly increased in patients treated with angiogenesis inhibitors [191]. On the other hand, the TIKs sorafenib and sunitinib were associated with hypertension, proteinuria, bleeding, skin reactions, hand–foot syndrome, fatigue, and diarrhea [192]. Remarkably, bevacizumab, the leader in clinical therapy to suppress tumor angiogenesis, showed significant side effects and drug resistance. The bevacizumab treatment was associated with proteinuria, hypertension, gastrointestinal perforation, hemorrhage, and stroke [190,192]. In addition, long-term treatment with bevacizumab may lead to the development of drug resistance due to the upregulation of other redundant tumor-derived angiogenic factors [190]. Furthermore, in most cancers, including breast, melanoma, pancreatic, and prostate cancer, bevacizumab failed to increase survival [19].

DA and DA-Ag are not without side effects, and DA-Ag has been observed to cause peripheral edema; orthostatic hypotension; hallucinations; and the sudden onset of sleeping and impulse-control disorders, including hypersexuality and compulsive eating, gambling, and shopping [17,193]. However, VEGF inhibitors have a higher and more severe side effect profile than DA and its DA-Ag, which can be accepted for certain cancer patients due to their critical situation but can certainly be improved for patients with endometriosis, OA, and some types of cancer [17,81]. In addition, the DA treatment significantly reduced the angiogenesis and growth of orthotopic HT29 colon cancer and subcutaneous Lewis lung carcinoma tumors; DA was also able to ameliorate neutropenia induced by commonly used anticancer drugs, and DA did not cause hypertension or hematologic, renal, or hepatic toxicity in normal mice, HT29-bearing mice, or Lewis lung carcinoma-bearing mice [83]. Additionally, DA or D2R DA-Ag could be a safer option in patients with or at risk for cardiovascular complications [17,83]. Treatment with a D2R agonist has been shown to block tumor growth, induce the regression of an aberrant blood supply, and normalize blood vessels in a mutant mouse model, and chronic treatment is able to restore the disturbed balance between proangiogenic and antiangiogenic factors [194]. Furthermore, cabergoline has been widely used and is considered a safe and nontoxic medication [195,196,197,198,199,200].

On the other hand, as we mentioned before, another major disadvantage of angiogenesis inhibitory agents, especially monoclonal antibodies, is the cost. For example, two studies estimated the average cost of an antiangiogenic therapy with bevacizumab, sorafenib, and sunitinib to be 13,500 USD, 6100 USD, and 6900 USD per patient per month, respectively [201,202]. However, for DA and DA-Ag, the costs vary between 300 USD and 600 USD per patient per month [17,193,203]. This means that DA treatment is ~10 to 45 times cheaper than angiogenesis inhibitors. This significant difference in cost provides an additional benefit to the antiangiogenic potential of DA.

However, although DA and DA-Ag have important advantages over monoclonal antibodies and TKIs in that they are economically more accessible, have greater availability, and probably have fewer adverse effects, large human studies are needed to clearly determine the full spectrum of the safety, dosing, and efficacy of using DA-related therapy as a new treatment in cancer, endometriosis, and OA. This could lead to lower costs and reduced adverse effects, which is the priority of the health care system.

## 8. Conclusions

The antiangiogenic effects of DA and DA-Ag have therapeutic potential for cancer, endometriosis, and OA, with potential advantages over angiogenic inhibitory monoclonal antibodies. However, further clinical studies are needed to demonstrate the efficacy and safety of DA and DA-Ag as adjuvants in pathologies requiring a reduction in angiogenesis. Further studies are needed to study which alternative proangiogenic pathways can be activated in the pathologies in which DA and DA-Ag are used as adjuvants. After this, it is possible to propose combinations of DA and DA-Ag with inhibitors of activated alternate proangiogenic pathways to increase their therapeutic efficacy.

## Figures and Tables

**Figure 1 ijms-24-10199-f001:**
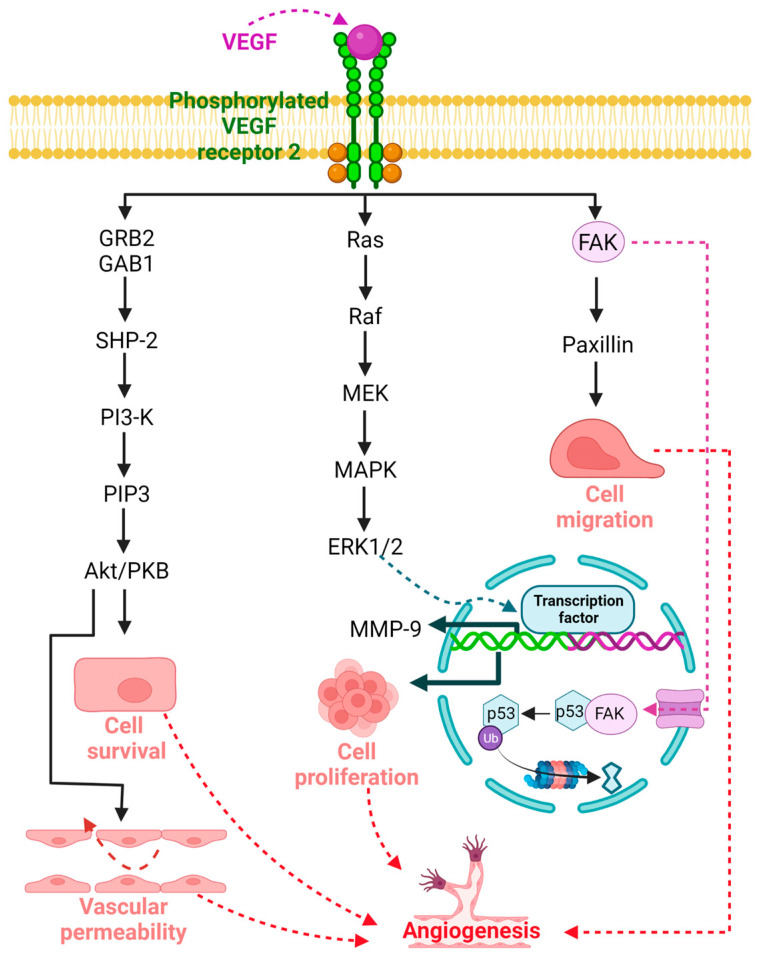
Signaling mediated by vascular endothelial growth factor (VEGF) and its target receptor VEGFR 2 plays an important role in angiogenesis under both physiological and pathological conditions. The signal transduction cascade begins when VEGF binds to VEGFR 2, leading to a conformational change, dimerization, and phosphorylation of tyrosine residues of the receptor, resulting in the activation of several intracellular molecules that act as downstream signaling elements involved in cell survival, vascular permeability, proliferation, and cell migration that promote angiogenesis.

**Table 1 ijms-24-10199-t001:** Experimental studies and/or clinical trials in which the effects of DA and DA-Ag on endometriosis are being studied.

Intervention	Model Analyzed	Observations
Cabergoline every three days at 50 µg/kg by oral gavage [3]	Heterologous mouse model	Cabergoline significantly decreased the lesion size, vascular density, and innervation in DA-Ag and anti-VEGF groups in comparison with control.
Cabergoline at 0.05 or 0.1 mg/kg/day orally for 14 days [81]	Implantation of human endometrium in female mice	The formation of new blood vessels was suppressed in endometriosis lesions, and a decrease in cellular proliferation was observed. VEGFR 2 phosphorylation was significantly lower in cabergoline-treated animals than controls.
Cabergoline at 0.05 (low dose) and 0.1 (high dose) mg/kg per day for 14 days [82]	Implantation of human endometrium (with mild or severe endometriosis) in female mice	D2R gene and protein expression was observed in human endometrial implants. Moreover, VEGF gene and VEGF and VEGFR 2 protein expressions were significantly lower in endometrial lesions treated with cabergoline than in controls.
Cabergoline 50 mg/kg per day orally or quinagolide 50 or 200 mg/kg per day for 14 days [95]	Nude mice with eutopic human endometrial fragments	Quinagolide and cabergoline both were effective at decreasing endometriotic lesion size and its cellular proliferation. Additionally, a reduction in VEGFR 2 and VEGF gene expression was observed.
Quinagolide at 200 μg/kg/day [102]	Wistar rats with Endometriosis was surgically induced by transplantation of autologous endometrial tissue	Quinagolide induced a significant regression in endometriotic implants and reduced the interleukin (IL)-6 and VEGF levels in peritoneal fluid.
Vaginal bromocriptine at a dose of 5 mg daily for 6 months of treatment [104]	Women with adenomyosis	Bromocriptine induced a significant improvement in menstrual bleeding and pain.
Cabergoline at 0.1 mg/kg/day by oral gavage for 4 weeks [106]	Autotransplantation of endometrial tissue on adult Sprague–Dawley rats	Cabergoline was not effective at endometriotic implant regression.
Cabergoline and Bromocriptine at 0.1 mg/kg/day orally for 30 days [107]	Induced endometriosis in Wistar rats	Bromocriptine and cabergoline significantly decreased the area (stromal and glandular tissue) of the endometriotic implants in comparison with controls.
Cabergoline at 0.5 mg/kg/day subcutaneously for 21 days [108]	Sprague–Dawley rats with endometriosis implantation	Cabergoline decreased the size and histopathological grade of the induced endometrial lesions.
Cabergoline at 0.075 mg/kg for 22 days [109]	Wistar rats with induction of endometriosis	Cabergoline produced a pronounced inhibitory effect on ectopic endometrioid formation.
Cabergoline at 0.05 mg/kg for 14 days [110]	C57BL/6 mice and ICR mice with induced endometriosis	Treatment with cabergoline diminished the inflammation in the uterus, peritoneum, and intestine in the recipient mice. Additionally, cabergoline decreased the expression pattern and localization of estrogen receptor beta (ER-β) and nerve growth factor (NGF).
Cabergoline 0.5 mg tablets, twice a week for 12 weeks [111]	Women with endometriosis	Cabergoline decreased the size of endometrioma.
Cabergoline 0.25 mg twice weekly for 6 months [112]	Women with endometriosis-associated pain syndrome	Cabergoline combined with hormone therapy standard schemes reduced the pain syndrome in patients with genital I–III endometriosis degree.
Quinagolide at 50 or 200 mg/kg per day for 14 days [113]	Women with endometriosis	Quinagolide induced a 69.5% reduction in the size of the lesions.
Cabergoline (0.5 mg twice weekly × 6 months) [114] Clinical trial NCT02542410. Status: completed	Women with endometriosis	In this pilot study, the change in the worst pain score (time frame: 6 months) after receiving cabergoline was measured. Moreover, changes in the sizes (mm) of endometrioma, deep infiltrating endometriosis, and adenomyosis lesions summed by type on magnetic resonance images at cycle 4 were also measured (time frame: at baseline and at menstrual cycle 4). Cabergoline decreased the pain score, and changes in the endometrial lesions inhibiting size and blood vessel growth was observed.
Quinagolide (1080 µg with daily target release rate of 13.5 µg) [115]. Clinical trial NCT03749109. Status: completed	Women with endometriosis	In this clinical trial, changes in the sizes (mm) of endometrioma, deep infiltrating endometriosis, and adenomyosis lesions (time frame: at baseline and at menstrual cycle 4) were measured via magnetic resonance at cycle 4. Quinagolide decreased the number and size of the endometrial and adenomyosis lesions.
Cabergoline (0.5 mg twice weekly for 6 months) [116]. Clinical trial NCT03928288. Status: recruiting	Women with endometriosis	In this clinical trial, the authors will measure changes in pain severity with different scales: the brief pain inventory interference scale (BPI), visual analog scale (VAS), and Biberoglu and Behrman patient ratings scale (B&B) over 6 months (time frame: every 6 weeks for 6 months).

**Table 2 ijms-24-10199-t002:** Experimental studies and/or clinical trials in which the effects of DA and DA-Ag on cancer are being studied.

Intervention	Model Analyzed	Observations
Cancer therapy using cobalt ferrite (CF) nanoparticles as a DA delivery agent by functionalizing CF-DA-polyethylene glycol (PEG) [125]	Human A549 cells lung cancer	CF-DA-PEG nanoparticles showed an anticancer effect by inducing apoptosis through activating the cytochrome-c and caspase-dependent apoptotic pathway and reactive oxygen species generation.
DA delivery via pH-sensitive nanoparticles [126]	Breast cancer mouse model	Nanoparticles induce tumor blood vessel normalization, improving the antitumor chemotherapeutic efficacy of doxorubicin.
DA 25 mg/kg twice a week [127]	Mouse model (C57BL/6) of pancreatic cancer	DA has synergistic roles with chemotherapy for pancreatic cancer by suppressing tumor-associated macrophages-derived inflammations.
Cabergoline (total week dose of 3.5 mg, starting 6 months after transsphenoidal surgery) [128]. Clinical trial NCT03271918. Status: completed	Subjects with pituitary adenoma	Tumor shrinkage, tumor rest stabilization, and cardiovascular safety (time frame: 24 months).
Cabergoline at a dose of 1 mg orally, twice a week for 4 weeks [129]	Women with breast cancer	Cabergoline was well tolerated, and although the overall response rate was low, a small subgroup of patients experienced prolonged disease control.
DA vasopressor dose individually titrated according to mean arterial pressure [130]. Clinical trial NCT02241083 Status: completed	Subjects with head and neck cancer	Evidence of clinically definite ischemia (time frame: 72 h).
Cabergoline, bromocriptine, or quinagolide (DA-Ag) [131]. Clinical trial NCT04107480. Status: recruiting	Subjects with prolactinoma	Health-related quality of life (time frame: 12 months) and long-term remission (time frame: 36 months).
Ropirinole (0.25 mg/day–6.0 mg/day oral) [132]. Clinical trial: NCT03038308. Status: Completed	Subjects with prolactinoma	Percentage of subjects that achieved stable PRL normalization (time frame: 6–12 months). A dose-dependent PRL nadir occurred 4.4 ± 1.2 h after drug intake, and PRL concentrations transiently normalized.
Cabergoline (twice weekly for weeks 1–4. Courses repeat every 4 weeks in the absence of disease progression or unacceptable toxicity) [133]. Clinical trial NCT01730729 Status: Completed	Women with breast cancer	Overall response rate at 2 months (time frame: after 8 weeks (2 cycles) of treatment).

**Table 3 ijms-24-10199-t003:** Experimental studies and/or clinical trials in which the effects of DA and DA-Ag in osteoarthritis (OA) are being studied.

Intervention	Model Analyzed	Observations
In vitro: DA 100 μM In vivo: DA was administered by intra-articular injection once a week for 12 weeks [139]	C28/I2 cells and primary cell culture of human chondrocytes Eight-week-old C57BL/6 male mice with surgically induced destabilization of the medial meniscus	In vitro, DA treatment inhibited the production of inducible nitric oxide synthase, COX-2, MMP-1, MMP-3, and MMP-13. DA reversed IL-1β-treated nuclear factor-kappa B activation and JAK2/STAT3 phosphorylation. In vivo, DA suppressed the degradation of cartilage matrix and reduced OA.
Copolymer P(DMA-co-MPC) with DA hydrochloride (5 g, 26.5 mmol) [183]	Mouse MC3T3-E1 osteoblastic cells	Improved lubrication and decreased reactive oxygen species in joint inflammation.
DA–melanin nanoparticles [184]	Primary chondrocytes were isolated from knee joint cartilage of 3-day-old Sprague–Dawley rats	DA–melanin nanoparticles have excellent anti-inflammatory and chondroprotective effects by inhibiting intracellular reactive oxygen species and reactive nitrogen species in vitro and in vivo.
Injectable hydrogel (alginate–DA, chondroitin sulfate, and regenerated silk fibroin) [185]	The adhesive strength of the material was measured by using a porcine skin interface and porcine cartilage ex vivo model	Hydrogel enhanced bone-marrow-derived mesenchymal stem cells recruitment, proliferation, and differentiation, as well as cartilage regeneration in a rat model.
D1R stimulation with fenoldopam D2R stimulation with ropinirole [186]	Cell culture and synovial fibroblasts from knee tissue from patients with rheumatoid arthritis and OA who underwent knee joint replacement surgery	Showed the involvement of the dopaminergic pathway in migration of synovial fibroblasts, supporting the therapeutic potential of the dopaminergic pathway in RA and in OA.

## Data Availability

Data sharing is not applicable.

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
