# Peer review of "Antiangiogenic Effect of Dopamine and Dopaminergic Agonists as an Adjuvant Therapeutic Option in the Treatment of Cancer, Endometriosis, and Osteoarthritis"

_ijms, 2023, doi:10.3390/ijms241210199_

Round 1
Reviewer 1 Report
The article "Antiangiogenic effect of Dopamine (DA) and Dopaminergic agonists as an adjuvant therapeutic option in the treatment of cancer, endometriosis, and osteoarthritis (OA)" is interesting. The authors explained that DA and DA-Ag have an antiangiogenic effect in the treatment of diseases such as cancer, endometriosis, and OA, and also the advantages over monoclonal antibodies. However, the authors need to revise a few points mentioned below.
1. Keywords: Please rearrange them in alphabetical order.
2. Table 1: The written style of intervention and outcomes are complex, each column has a different style, please revise the contents and write in a simple way.
3. Table 2: Please mention the clinical trials identification number (NCT no.) and expected completion or phase or completion date.
4. The article needs to be more improvised to enhance its impact. Overall several grammatical and incomplete sentences have been found. Please revise/rewrite.
Author Response
The article "Antiangiogenic effect of Dopamine (DA) and Dopaminergic agonists as an adjuvant therapeutic option in the treatment of cancer, endometriosis, and osteoarthritis (OA)" is interesting. The authors explained that DA and DA-Ag have an antiangiogenic effect in the treatment of diseases such as cancer, endometriosis, and OA and advantages over monoclonal antibodies. However, the authors need to revise a few points mentioned below.
- Keywords: Please rearrange them in alphabetical order.
Response: The order of keywords was modified as you suggested.
- Table 1: The written style of intervention and outcomes are complex, each column has a different style, please revise the contents and write in a simple way.
Response: The style and contents of Table 1 were totally modified.
- Table 2: Please mention the clinical trials identification number (NCT no.) and expected completion or phase or completion date.
Response: The NCT number was added to Tables 1 and 2, and the style and content were modified.
- The article needs to be more improvised to enhance its impact. Overall several grammatical and incomplete sentences have been found. Please revise/rewrite.
Response: We appreciate your valuable comments. We have carefully reviewed and corrected the grammatical mistakes as you suggested.
Reviewer 2 Report
Reviewer comments and suggestions
The review reported the mechanisms of antiangiogenic action of the Dopamine (DA)- dopaminergic D2 receptor D2R/VEGF-VEGFR 2 system, and to summarizes the data-related findings in biomedical and clinical trials in cancer, endometriosis, and osteoarthritis (OA). Articles explaining the antiangiogenic effect of DA and DA-Ag, research articles, reviews and clinical trials were considered. Finally, the authors highlighted that DA and DA-Ag may have antiangiogenic effects that could reinforce the treatment of diseases such as cancer, endometriosis, and OA.
Overall, the manuscript was well written. However, a few concerns/comments needed to be explained/modified.
- Line 36-37 What does it mean ": reduction of blood vessel formation is one of the most important therapeutic approaches.
- Line 40-41 Method section should be well explained.
- Line 57 (reference 5-13) is not an appropriate way to represent. Its better to add one or two references.
- Line 68-70 The introduction needs to add more points.
- Line 72 typoerror (Metodology)
- Line 74-75 Please mention the years.
- Line 107-116 figures are preferable to text.
- Line 144-145 Is it important to cite so many references.
- The figures were originally prepared or taken from any other study ( figure 2)
- Please avoid long sentences 252-255
- Line 256-257 The authors could describe it well.” relevance has been 256 the use of monoclonal antibodies targeting the VEGF pathway for cancer therapy [99,100].”
- Please check the outcome of Table 1 “Cabergoline (0.5 mg twice weekly for 6 months) [116] Women with endometriosis”
- Line 466-476 please mention the clinical trial data.
- Please check the guidelines of MDPI, it seems that the authors need to modify all the references.
Author Response
This review reports the mechanisms of antiangiogenic action of the dopamine (DA)-dopaminergic D2 receptor D2R/VEGF-VEGFR 2 system and summarizes the data-related findings in biomedical and clinical trials in cancer, endometriosis, and osteoarthritis (OA). Articles explaining the antiangiogenic effect of DA and DA-Ag, research articles, reviews and clinical trials were considered. Finally, the authors highlighted that DA and DA-Ag may have antiangiogenic effects that could reinforce the treatment of diseases such as cancer, endometriosis, and OA.
Overall, the manuscript was well written. However, a few concerns/comments needed to be explained/modified.
- Line 36-37 What does it mean ": reduction of blood vessel formation is one of the most important therapeutic approaches.
Response: “reduction of blood vessel formation could be one of the most important therapeutic approaches”. However, due to the limitation of the number of words in the abstract, this sentence was removed.
- Line 40-41 Method section should be well explained.
Response: The method section was modified.
- Line 57 (reference 5-13) is not an appropriate way to represent. It is better to add one or two references.
Response: This text was modified.
- Line 68-70 The introduction needs to add more points.
Response: We added more information in the introduction as you suggested.
- Line 72 typoerror (Metodology)
Response: The typographic error was corrected.
- Line 74-75 Please mention the years.
Response: The years were included.
- Line 107-116 figures are preferable to text.
Response: We agree with you; however, a figure was not included for this section because it was already designed and published in the Bandala 2023 article. We added the citation in the text (reference 34).
- Line 144-145 Is it important to cite so many references.
Response: We prefer to cite each of the studies since different dopaminergic drugs were evaluated for the treatment of Parkinson's disease.
- The figures were originally prepared or taken from any other study (figure 2).
Response: Both figures are original. Figure 2 was designed to analyze and integrate the information from the corresponding studies cited in the text and in the figure caption.
- Please avoid long sentences 252-255
Response: The sentence was modified.
- Line 256-257 The authors could describe it well.” relevance has been 256 the use of monoclonal antibodies targeting the VEGF pathway for cancer therapy [99,100].
Response: The sentence was modified.
- Please check the outcome of Table 1 “Cabergoline (0.5 mg twice weekly for 6 months) [116] Women with endometriosis”
Response: The outcome of the clinical trial was revised and corrected.
- Line 466-476 please mention the clinical trial data
Response: None of these cited studies correspond to clinical trial data; however, we added “mutant mouse model” in reference 194 for better explanation, and we added 5 citations of articles developed in humans to support the sentence “Furthermore, cabergoline has been widely used and is considered a safe and nontoxic medication”
- Please check the guidelines of MDPI, it seems that the authors need to modify all the references.
Response: The format of references was corrected.
Reviewer 3 Report
Hi !
Good work. Very useful for the scientific domain (people working in the field)
Reading this paper once more, you will find a few grammar issues.
Author Response
Good work. Very useful for the scientific domain (people working in the field)
Response: We appreciate your comment, and hope that this review can be useful to promote more research on the matter.
Reading this paper once more, you will find a few grammar issues.
Response: We have carefully reviewed and corrected the grammatical errors as you suggested.
Reviewer 4 Report
Antiangiogenic Effect of Dopamine and Dopaminergic 2 Agonists as an Adjuvant Therapeutic Option in the Treatment 3 of Cancer, Endometriosis, and Osteoarthritis
General Comments: The review of Torreblanca et al is very interesting. It discusses the antiangiogenic mechanisms of Dopamine and its antagonists and their therapeutic capability as adjuvant in cancer, endometriosis, and OA. The authors also discuss the advantages and disadvantages of Dopamine and its antagonists compared to VEGF/VEGFR 2 monoclonal antibodies. The manuscript is well organized and clearly transcribed. The Title and Abstract are illustrative of the study. The objective is clearly indicated and structurally line up with the methods and results along with strong and informative figures supporting the conclusion. However, the following concerns may be addressed before acceptance:
1. Line 72, Spelling of Methodology may be corrected.
2. Most of the studies supporting the data needs additional clinical confirmation through other statistical studies. Also, authors may please consider adding source of information in the legend of each figure.
Author Response
General Comments: The review of Torreblanca et al is very interesting. It discusses the antiangiogenic mechanisms of dopamine and its antagonists and their therapeutic capability as adjuvants in cancer, endometriosis, and OA. The authors also discuss the advantages and disadvantages of dopamine and its antagonists compared to VEGF/VEGFR 2 monoclonal antibodies. The manuscript is well organized and clearly transcribed. The Title and Abstract are illustrative of the study. The objective is clearly indicated and structurally line up with the methods and results along with strong and informative figures supporting the conclusion. However, the following concerns may be addressed before acceptance:
- Line 72, Spelling of Methodology may be corrected.
Response: We corrected the grammatical mistakes in the Methodology section as you suggested.
- Most of the studies supporting the data need additional clinical confirmation through other statistical studies.
Response: We agree with your comment, and more clinical studies with solid statistical analysis are required to evaluate the efficacy of dopaminergic drugs, especially in endometriosis and osteoarthritis. One of the objectives of this review was to arouse the interest of researchers to advance the knowledge that has been based on in vitro and in vivo studies.
- Additionally, authors may please consider adding source of information in the legend of each figure.
Response: It is important to note that both figures are original. In the case of figure 1, the information is general, figure 2 was designed by analyzing and integrating information from the corresponding studies cited in the text and in the figure caption.
Round 2
Reviewer 1 Report
The authors made changes as per the comments.
Minor grammatical/spacing errors to be corrected.